# Training Stronger Baselines for Learning to Optimize

**Tianlong Chen[1*], Weiyi Zhang[2*], Jingyang Zhou[3], Shiyu Chang[4],**
**Sijia Liu[4], Lisa Amini[4], Zhangyang Wang[1]**
[1]University of Texas at Austin, [2]Shanghai Jiao Tong University,
[3]University of Science and Technology of China, [4]MIT-IBM Watson AI Lab, IBM Research
{tianlong.chen,atlaswang}@utexas.edu,{weiyi.zhang2307,djycn1996}@gmail.com
{shiyu.chang,sijia.liu,lisa.amini}@ibm.com

## Abstract

Learning to optimize (**L2O**) is gaining increased attention because classical optimizers require laborious, problem-specific design and hyperparameter tuning. However, there are significant performance and practicality gaps between manually designed optimizers and existing L2O models. Specifically, learned optimizers are applicable to only a limited class of problems, often exhibit instability, and generalize poorly. As research efforts focus on increasingly sophisticated L2O models, we argue for an orthogonal, under-explored theme: improved training techniques for L2O models. We first present a progressive, curriculum-based training scheme, which gradually increases the optimizer unroll length to mitigate the well-known L2O dilemma of truncation bias (shorter unrolling) versus gradient explosion (longer unrolling). Secondly, we present an off-policy imitation learning based approach to guide the L2O learning, by learning from the behavior of analytical optimizers. We evaluate our improved training techniques with a variety of state-of-the-art L2O models and immediately boost their performance, *without making any change to their model structures*. We demonstrate that, using our improved training techniques, **one of the earliest and simplest L2O models [1] can be trained to outperform even the latest and most complex L2O models** on a number of tasks. Our results demonstrate a greater potential of L2O yet to be unleashed, and prompt a reconsideration of recent L2O model progress. Our codes are publicly available at: `https://github.com/VITA-Group/L2O-Training-Techniques`.

## 1 Introduction

Learning to optimize (L2O) [1–10], a rising subfield of meta learning, aims to replace manually designed analytical optimizers with learned optimizers, i.e., update rules as functions that can be fit from data. An L2O method typically assumes a **model** to parameterize the target update rule. A trained L2O model will act as an algorithm (*optimizer*) itself, that can be applied to training other machine learning models, called *optimizees*, sampled from a specific class of similar problem instances. The training of the L2O model is usually done in a meta-fashion, by enforcing it to decrease the loss values over sampled optimizees from the same class, via certain **training techniques**.

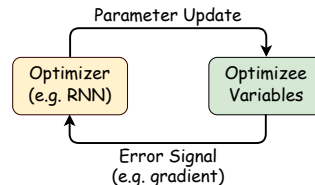

Parameter Update

Figure 1: Learning to optimize.

Earlier L2O methods refer to black-box hyper-parameter tuning approaches [11–16] but these methods often scale up poorly [17] when the optimizer's parameter amounts grow large. The recent mainstream works in this vein [1, 6–9] leverage a recurrent network as the L2O model, typically long short-term memory (LSTM). That LSTM is **unrolled** to mimic the behavior of an iterative optimizer and trained

---

[*]Equal Contribution.

to fit the optimization trajectory. At each step, the LSTM takes as input some optimizee's current measurement (such as zero- and first-order information), and returns an update for the optimizee. A high-level overview of L2O workflow is presented in Figure 1.

Although learned optimizers outperform hand-designed optimizers on certain problems, they are still far from being mature or practically popular. Training LSTM-based L2O models is notoriously difficult and unstable; meanwhile, the learned optimizers often suffer from poor generalization. Both problems arise from decisions made on the unroll length of the LSTM during L2O training [17–20]. For training modern models, an optimizer (either hand-crafted or learned) can take thousands of iterations, or more. Since naively unrolling the LSTM to this full length is impractical, most LSTM-based L2O methods [1, 6–9, 17, 21, 22] take advantage of truncating the unrolled optimization. As a result, the entire optimization trajectory is divided into consecutive shorter pieces, where each piece is optimized by applying a truncated LSTM. However, choosing the unrolling (or division) length faces a well-known dilemma [7]: on one hand, a short-truncated LSTM can result in premature termination of the iterative solution. The resulting "truncation bias" causes learned optimizers to exhibit instability and yield poor-quality solutions when applied to training optimizees. On the other hand, although a longer truncation is favored for optimizer performance, if unrolled too long, common training pitfalls for LSTM, such as gradient explosion, will result [19].

To tackle these challenges, a number of solutions [7–9, 17] have been proposed, mainly devoted to designing more sophisticated architectures of L2O models (LSTM variants), as well as enriching the input features for L2O models [7–9]. These approaches introduce complexity, despite continuing to fall short on performance, generalization, and stability issues. In contrast, this paper seeks to improve L2O from an orthogonal perspective: *can we train a given L2O model better*?

**Our Contributions**    We offer a toolkit of **novel training techniques**, and demonstrate that, solely by introducing our techniques into the training of existing, state-of-the-art L2O models, our methods substantially reduce training instability, and improve performance and generalization of trained models. More specifically:

- We propose a progressive training scheme to gradually unroll the learned optimizer, based on an exploration-exploitation view of L2O. We find it to effectively mitigate the dilemma between truncation bias (shorter unrolling) v.s. gradient explosion (longer unrolling).

- We introduce off-policy imitation learning to further stabilize the L2O training, by learning from the behavior of hand-crafted optimizers. The L2O performance gain by imitation learning endorses the implicit injection of useful design knowledge of good optimizers.

- We report extensive experiments using a variety of L2O models, optimizees, and datasets. Incorporating our improved training techniques immediately boosts all state-of-the-art L2O models, without any other change. We note especially that the earliest and simplest LSTM-based L2O baseline [1] can be trained to outperform latest and much more sophisticated L2O models.

Our results offer an important reminder when tackling new domains, such as L2O, with machine learning techniques: besides developing more complicated models, training existing simple baselines better is equally important. Only by fully unleashing "simple" models' potential, can we lay a solid and fair ground for evaluating the L2O progress.

## 2   Related Work

**Learning to Optimize**    L2O uses a data-driven learned model as the optimizer, instead of hand-crafted rules (e.g., SGD, RMSprop, and Adam). [1] was the first to leverage an LSTM as the coordinate-wise optimizer, which is fed with the optimizee gradients and outputs the optimizee parameter updates. [6] instead took the optimizee's objective value history, as the input state of a reinforcement learning agent, which outputs the updates as actions. To train an L2O with better generalization and longer horizons, [7] proposes random scaling and convex function regularizers tricks. [8, 23] introduce a hierarchical RNN to capture the relationship across the optimizee parameters and trains it via meta learning on the ensemble of small representative problems. Besides learning the full update rule, L2O was also customized to automatic hyperparamter tuning in specific tasks [24–26].

**Curriculum Learning**    The idea [27] is to first focuses on learning from a subset of simple training examples, and gradually expanding to include the remaining harder samples. Curriculum learning

often yields faster convergence and better generalization, especially when the training set is varied or noisy. [28] unifies it with self-paced learning. [29] automates the curriculum learning by employing a non-stationary multi-armed bandit algorithm with a reward of learning progress indicators. [30–33] describe a number of applications where curriculum learning plays important roles.

**Imitation Learning**    Imitation learning [34, 35], also known as "learning from demonstration", is to imitate an expert demonstration instead of learning from rewards as in reinforcement learning. In our work, L2O imitates multiple hand-crafted experts, such as Adam, SGD, and Adagrad. A relevant work is the "Lookahead Optimizer" [36], which first updates the "fast weights" $k$ times using a standard optimizer in its inner loop before updating the "slow weights" once. Another work of self-improving [37, 38] chooses several optimizers for hybrid training, including the desired L2O and several other auxiliary (analytical) optimizers, per a probability distribution. It then gradually anneals the probability of auxiliary optimizers and eventually leaves only the target L2O model in training.

# 3    Strengthening L2O Training with Curriculum Learning

**Problem Setup for L2O**    The goal is to learn an optimizer parameterized by $\phi$, which is capable of optimizing a similar class of optimizee functions $f(\theta)$. In the case of machine learning, $\theta$ is the model parameter and $f(\theta)$ is the objective loss. Usually, the learned optimizer outputs $\Delta\theta_t$ and the optimizee parameter is updated as $\theta_{t+1} = \theta_t + \Delta\theta_t$. The inputs of the learned optimizer can be $\nabla f(\theta_t)$ or other optimizee features. The learned optimizer is trained by a loss $\mathcal{L}_f(\phi) = \sum_{t=1}^{\mathrm{N_{train}}} \omega_t f(\theta_t; \phi)$ where $\mathrm{N_{train}}$ is called the horizon of optimization trajectory (i.e., the unrolling length).

## 3.1    An exploration-exploitation view for L2O

Training an L2O model follows the basic idea of meta learning, by applying the L2O model to training a number of sampled optimizees, and enforcing it to decrease their loss values as much as possible. We refer the readers to [1] for the standard training setting in details.

During training L2O, we recognize that the the truncated optimization (e.g., unrolling LSTM to a limited number of iterations) is one of the main culprits, for the instability and poor quality when applying trained L2O to optimizees. That is because for each update, L2O can only learn from a short segment of optimization trajectory, so that it sacrifices generalization ability to longer horizons [7].

One plausible remedy for alleviating this truncation bias is to augment L2O training with more optimizees, as well as longer optimization trajectory for each optimizee. This can be interpreted as addressing an exploration-exploitation problem, a typical challenge in reinforcement learning [39]. On one hand, the agent (L2O) has to exploit policies (i.e., optimization trajectory for each optimizee currently being trained) that are previously better rewarded, in hope to gain more rewards at present. On the other hand, the same L2O agent also needs to explore unfamiliar state-action choices (here more optimizees) to discover other potentially (more) favorable policies.

**Exploration: training with more optimizees**    A good learned optimizer needs to explore and see various optimizees' landscapes, in order to generalize to tackle unseen optimizees with more diverse landscapes. More exploration could be considered as a special form of data augmentation, which was found beneficial for L2O training [7].

One straightforward option is sampling more optimizees. In practice, we find an even easier-to-implement alternative to be the same effective: we re-use sampled optimizees, by starting from their different random weight initializations. It is similar to the "exploring starts" in reinforcement learning. Note that this also artificially enlarges our meta-training set and thus calls for meta-epochs.

**Exploitation: training longer for each optimizee**    A good learned optimizer also shall aim to be fully executed, so as to reach low optimizee loss and high-precision solution. That is especially important for training the L2O model to see the landscapes close to the minimum. Moreover, just like any other iterative optimizer, a learned optimizer predicts its current update from the past history, and therefore should also benefit from more training exposure to longer-term dependency.

In view of the above two, the first (basic) improvement we take is to simply train the L2O model with more optimizees, and longer optimization trajectories for each. Naive as it might look like, in section 5.1 we will show that they clearly benefit existing L2O models, and solidify our starting point.

## 3.2 Curriculum learning to adaptively balance exploration and exploitation

Inspired by the exploration-exploitation view, we next introduce a new training strategy based on curriculum learning [27], for the first time to L2O models. The curriculum is defined by gradually unrolling the optimizer more, w.r.t. the number of epochs. Specifically, we denote the number of training unrolling steps by $N_{train}$, and the number of validation unrolling steps by $N_{valid}$. We progressively increase $N_{train}$ and $N_{valid}$ during training, until the validation loss stops to decrease.

Two things that need to be determined for the curriculum are: how to grow $N_{train}/N_{valid}$, and when to stop. Specifically, we set a sequence of $N_{train} = [N_{train}^0, N_{train}^1, ...]$, e.g., $N_{train} = [100, 200, 500, 1000, ...]$ by default. To alleviate *overfitting on unrolling length* during L2O training, we intentionally choose *mismatched unrolling steps* in validation, with $N_{valid}^i = N_{train}^{(i+1)}$ as our default case, e.g, $N_{valid} = [200, 500, 1000, ...]$ to evaluate if the L2O model generalizes to longer horizons.

In our curriculum learning schedule, we validate the learned optimizer every $T_{period}$ training epoch, where a single epoch indicates an optimization trajectory of $N_{train}$ steps. We set a minimum number $N_{period}$ of training periods. Within each period, we train the L2O model for $T_{period}$ epochs. For each $N_{train}^i$, we train at least $N_{period}$ periods and keep the model of the lowest validation loss. We continue to train L2O with $N_{train}^i$, if the last period's validation loss is the lowest during the $i^{th}$ training stage, otherwise we switch to the next training stage. Before starting training with $N_{train}^{(i+1)}$, we first validate the previous best model with $N_{valid}^{(i+1)}$ as the baseline validation loss of the $(i+1)^{th}$ training stage. If none of the $N_{period}$ validation loss in the $(i+1)^{th}$ training stage is lower than the baseline loss, we stop training and export the best model of the $i^{th}$ training stage, as our final trained L2O model. The overall procedure is summarized in Algorithm 1.

---

**Algorithm 1:** Curriculum Learning for Training Learnable optimizer (L2O)

**Input:** $N_{train}$, $N_{valid}$, $N_{period}$, $T_{period}$, best validation loss $\mathcal{L}_{min}$ and current validation loss $\mathcal{L}_{val}$, index of training stage $i$, an L2O $\phi = \phi' = \phi_0$

**Output:** An updated L2O $\phi$

1  **while** True **do**
2     $n = 1$, *stop* = True ;
3     **while** $n \leq N_{period}$ *or* $\mathcal{L}_{val} == \mathcal{L}_{min}$ **do**
4       $n = n + 1$;
5       **for** $t = 1, 2, ..., T_{period}$ **do**
6         Update $\phi'$ a epoch with $N_{train}^i$;
7       **end**
8       $\mathcal{L}_{val}$ =val. loss with $\phi'$ and $N_{valid}^i$;
9       **if** $\mathcal{L}_{min} > \mathcal{L}_{val}$ **then**
10        $\mathcal{L}_{min} = \mathcal{L}_{val}$, $\phi = \phi'$,
11        *stop* = False;
12      **end**
13    **end**
14    **if** *stop* == True **then**
15      break;
16    **end**
17    $i = i + 1$, $\phi' = \phi$;
18    $\mathcal{L}_{min}$ =val. loss with $\phi$ and $N_{valid}^i$;
19 **end**

---

**Algorithm 2:** Imitation Learning for L2O

**Inputs:** L2O $\phi$ and analytical optimizers $(\mathcal{O}_i)_{i \in I}$, a threshold $r$, $T_{total}$, $N_{train}$

**Output:** An updated L2O $\phi$

1  **for** $t = 1, 2, ..., T_{total}$ **do**
2     Sample a number from uniform distribution $u \sim \mathcal{U}(0, 1)$;
3     **if** $u < r$ **then**
4       Select an optimizer $\mathcal{O}$ from $(\mathcal{O}_i)_{i \in I}$ with equal probabilities;
5       Generate an optimization trajectory $\mathcal{T} = [(\boldsymbol{g}_1, \Delta\boldsymbol{\theta}_1^{\mathcal{O}}), \cdots, (\boldsymbol{g}_{N_{train}}, \Delta\boldsymbol{\theta}_{N_{train}}^{\mathcal{O}})]$;
6       Update $\phi$ with respect to $\mathcal{L}_{\mathcal{O}}(\phi) = \sum_{t=1}^{N_{train}} \omega_t (\Delta\boldsymbol{\theta}_t^{\mathcal{O}} - \Delta\boldsymbol{\theta}_t^{\phi})^2$ ;
7     **end**
8     **else**
9       Generate an optimization trajectory by L2O $\phi$;
10      Update $\phi$ with respect to $\mathcal{L}_f(\phi) = \sum_{t=1}^{N_{train}} \omega_t f(\boldsymbol{\theta}_t; \phi)$;
11    **end**
12 **end**

---

The above curriculum training naturally enforces an adaptive balance between exploration and exploitation. We begin L2O training with a small number of optimizees sampled (exploration), and each trained with small unrolling steps (exploitation). As the exploration grows (more training

epochs), each optimizee is simultaneously exploited more, with a larger number of steps unrolled for optimization. In this way, we effectively avoid long chaotic trajectories generated by optimizers that have not been trained sufficiently at the beginning. Moreover, the optimizer can focus on learning simpler, short-horizon trajectory patterns firsts, and gradually grow to covering longer dependency, without overfitting difficult corner-cases. It is shown to alleviate the intrinsic bias caused by truncated optimization at different L2O training phases. As a result, the L2O model converges quickly and more stably, and ends up reaching lower validation losses.

## 4 Off-Policy Imitation Learning via Analytical Optimizers

### 4.1 Imitation learning: multi-task regularization by analytical optimizers

In this section, we propose another L2O training method based on imitation of analytical optimizers behaviours, through a multi-task learning form, which is found to further stabilize our training, prevent overfitting, and improve the trained L2O models' generalization.

The learned optimizer may "overfit" some training optimizees if it learns some optimization policies that are only applicable to some but not generalize to more unseen optimizees. To enforce a generalizability constraint, we regularize our L2O training by imitating some general optimization policies, which are known to generalize well and agnostic to data. For this purpose, we let the L2O model follow optimization trajectories generated by analytical optimizers such as Adam [40]. That imitation also help prevent the LSTM model's "spiking" or other highly non-smooth update predictions sometimes, stabilizing the optimization trajectories.

Generally, the learned optimizer outputs the optimizee parameter updates $\Delta \boldsymbol{\theta}_t$ and use $\boldsymbol{\theta}_{t+1} = \boldsymbol{\theta}_t + \Delta \boldsymbol{\theta}_t$ to calculate the optimizee $f(\boldsymbol{\theta}_{t+1})$. The optimizer is trained by a loss $\mathcal{L}_f(\boldsymbol{\phi}) = \sum_{t=1}^{N_{\text{train}}} \omega_t f(\boldsymbol{\theta}_t; \boldsymbol{\phi})$ on an optimization trajectory of length T generated by the current learned optimizer $\boldsymbol{\phi}$. Now given another analytical optimizer $\mathcal{O}$ such as Adam optimizer [40] as a "teacher", we first use $\mathcal{O}$ to generate an optimization trajectory of length $N_{\text{train}}$ on the training optimizee $f$. The trajectory is denoted by $\mathcal{T} = [(\boldsymbol{g}_1, \Delta \boldsymbol{\theta}_1^{\mathcal{O}}), (\boldsymbol{g}_2, \Delta \boldsymbol{\theta}_2^{\mathcal{O}}), ..., (\boldsymbol{g}_{N_{\text{train}}}, \Delta \boldsymbol{\theta}_{N_{\text{train}}}^{\mathcal{O}})]$ where $\boldsymbol{g}_t = \nabla f(\boldsymbol{\theta}_t)$. The relation between the inputs $(\boldsymbol{g}_t)_{t \in [1, N_{\text{train}}]}$ and the outputs $(\Delta \boldsymbol{\theta}_t^{\mathcal{O}})_{t \in [1, N_{\text{train}}]}$ is determined by the optimization policies of $\mathcal{O}$. Now we calculate another training objective which is the weighted square error between the label updates $(\Delta \boldsymbol{\theta}_t^{\mathcal{O}})_{t \in [1, N_{\text{train}}]}$ given by $\mathcal{O}$ and the updates given by the current learned optimizer $\boldsymbol{\phi}$ as $(\Delta \boldsymbol{\theta}_t^{\phi} = \phi(\boldsymbol{g}_t))_{t \in [1, N_{\text{train}}]}$ as shown in equation (1):

$$\mathcal{L}_{\mathcal{O}}(\boldsymbol{\phi}) = \sum_{t=1}^{N_{\text{train}}} \omega_t (\Delta \boldsymbol{\theta}_t^{\mathcal{O}} - \Delta \boldsymbol{\theta}_t^{\phi})^2 \tag{1}$$

Note that it is analogous to off-policy learning [41] in reinforcement learning where the agent is trained on a trajectory generated by another behavior policy. In our case, the learned optimizer is trained on the trajectory generated by the typically more stable analytical optimizer $\mathcal{O}$. We adjust the weight between the two losses $\mathcal{L}_f$ and $\mathcal{L}_{\mathcal{O}}$ for multi-task learning as shown in algorithm 2. Note that in reinforcement learning, off-policy is usually aimed at boosting exploration, while our goal differs here, i.e., to stabilize L2O training learn more generalizable optimizers.

### 4.2 Imitation Learning versus Self Improving

A self-improving approach [37, 38] was lately adapted for L2O training, by sampling and mixing different optimizers in one training pass (i.e., each iteration may adopt a different optimizer's update), according to a probability distribution. Then, the probability of choosing other optimizers will be gradually annealed, so that only the desired optimizer will remain at the end. Compared with imitation learning, self-improving technique only produce a single optimization trajectory $\mathcal{T}_s$ which consists of mixture update from different optimizers. Following the notation in the Section 4, the generated trajectory is presented as $\mathcal{T}_s = [(\boldsymbol{g}_1, \Delta \boldsymbol{\theta}_1^{\mathcal{O}_{i_1}}), (\boldsymbol{g}_2, \Delta \boldsymbol{\theta}_2^{\mathcal{O}_{i_2}}), ..., (\boldsymbol{g}_{N_{\text{train}}}, \Delta \boldsymbol{\theta}_{N_{\text{train}}}^{\mathcal{O}_{i_{\text{train}}}})]$ where $\boldsymbol{g}_t = \nabla f(\boldsymbol{\theta}_t)^1$ and $(\Delta \boldsymbol{\theta}_t^{\mathcal{O}_{i_t}})_{t \in [1, N_{\text{train}}]}$ are the corresponding update rule. Let $\{\mathcal{O}_0, \cdots, \mathcal{O}_k\}$ is the

candidate pool of optimizers, where $\mathcal{O}_0$ is the desired L2O and $\{\mathcal{O}_1, \cdots, \mathcal{O}_k\}$ are the auxiliary optimizers. At the beginning, $\{\mathcal{O}_{i_k}\}_1^{\text{train}}$ are sampled from $\{\mathcal{O}_0, \cdots, \mathcal{O}_k\}$ respect to a multinomial distribution $\mathcal{M}(p_0, p_1, \cdots, p_k)$, $p_0 = p_1 = \cdots = p_k = \frac{1}{k+1}$. Then, along with the increase of training epochs, $p_1 = \cdots = p_k$ decay to zero and $p_0$ grow to $1$. In this way, the desired L2O is steadily improved with the assistance of auxiliary optimizers.

## 5 Experiments and Analysis

In this section, we conduct systematic experiments to evaluate our proposed training techniques. Our experiments are organized into three parts, respectively showing:

- The earliest and simplest LSTM-based L2O baseline [1] can surpass some latest sophisticated L2O models, by training with our proposed techniques.
- Our improved training techniques can be further plugged into previous state-of-the-art L2O methods and yield extra performance boosts for them all.
- All the components in our proposal matter, i.e., an extensive ablation study for validating the respective gain of each technique.

**Optimizee Settings** For the fair comparison purpose, in our all experiments, we train the optimizer on the same single optimizee as in [1], which uses the cross-entropy loss on top of a simple Multi-layer Perceptron (MLP) with one hidden layer of 20 dimensions and the sigmoid activation function on the MNIST dataset. Validation also follows to use the same optimizee as in [1]. We consider five optimizee problems during testing, to evaluate the generalization ability of the learned optimizer:

- i) MLP-orig: the same MLP used for training on the MNIST;
- ii) MLP-ReLU: the previous MLP with activation function replaced by ReLU on the MNIST;
- iii) MLP-deeper: the previous MLP with 2 hidden layers of 20 dimensions on the MNIST.
- iv) Conv-MNIST: A convolutional neural network (CNN) with 2 convolution layers, 2 max pooling layers and 1 fully connected layer on the MNIST dataset. The first convolution layer uses $16$ $3 \times 3$ filters with stride $1$. The second convolution layers use $32$ $5 \times 5$ filters with stride $1$. The max pooling layers are of size $2 \times 2$ with stride $2$;
- v) Conv-CIFAR: the above CNN trained on CIFAR-10, with all identical architecture configurations except using stride $2$.

Optimizee i) is designed for sanity check and to exclude randomness factors. Optimizees ii)-iv) are for evaluating the generalization of L2O across network architectures. Finally, optimizee v) evaluates the generalization of L2O across both network architectures and datasets.

**Optimizer Settings** We take the earliest and simplest L2O-DM introduced by DeepMind [1] as a baseline, which has been long treated as a "poor-generalization baseline" [7,8]. Other two state-of-the-art (SOTA) L2Os with complicated architectures are examined too: RNNprop [7], and L2O-Scale [8]. Another variant of L2O-Scale in [8] is considered as another strong baseline, L2O-Scale-Meta[2].

**Training Details and Evaluation** L2O are trained by default meta optimizers with the best hyperparameters provided by each baseline: L2O-DM [1] and RNNprop [7] are optimized by Adam with an initial learning rate of $1 \times 10^{-3}$; L2O-Scale [8] and its variants are optimized by RMSProp with an initial learning rate of $1 \times 10^{-6}$. The optimizee parameters are initialized by a random normal distribution of standard deviation of $0.01$. The batch size for optimizees is $128$. The unroll length is fixed to $20$ except for the meta learning baseline of L2O-Scale where both the number of optimization steps and unroll lengths are sampled from long-tail distributions [8]. In the testing phase of L2O, we evaluate learned optimizers on unseen testing optimizees for $10,000$ steps every time, and then **report the training loss of testing optimizees**. Other hyperparameters are strictly controlled to be fair. Our results come with multiple independent runs, and the error bars are reported in the Appendix.

### 5.1 Training the L2O-DM baseline to surpass the state-of-the-arts

In this section, we equip the "simple baseline" L2O-DM with our improved training techniques, including the curriculum learning scheduler (CL) and the imitation learning regularizer (IL). L2O-

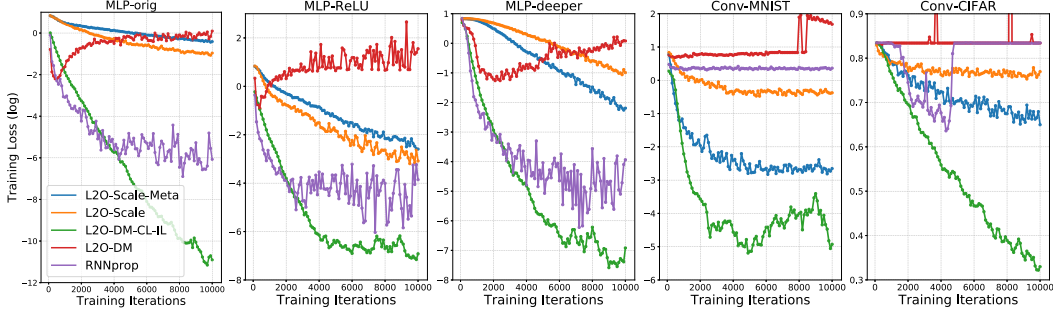

Figure 2: Evaluation performance of our enhanced L2O and previous SOTAs (i.e., log loss over unseen optimizees with learned L2O v.s. training iterations of optimizees). Each curve is the average of ten runs.

DM-CL-IL denotes the enhanced L2O model. Learned optimizers are evaluated on five representative optimizees and the corresponding optimizee training loss are collected in Figure 2.

From the results in Figure 2, we observe that the previously noncompetitive L2O-DM, that initially even cannot stably converge on the Optimizee i) at long horizons, now consistently and largely outperforms over all previous SOTA L2O methods: RNNprop, L2O-Scale, and L2O-Scale-Meta, by decreasing the objective loss value much lower. Compared to the vanilla L2O-DM, CL and IL are also observed to mitigate the large variances of the optimizee's training loss, especially in MLP-orig and MLP-ReLU cases. It confirms that our improved training techniques improve the L2O generalization and alleviate the training instability.

## 5.2 Training state-of-the-art L2O models to boost more performance

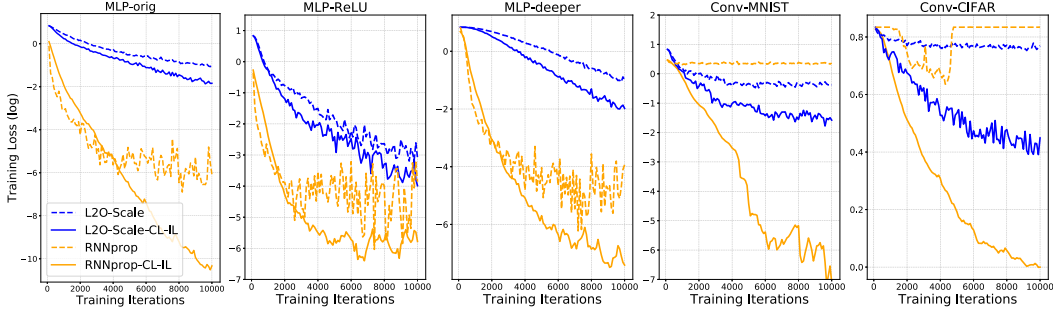

Figure 3: Evaluation comparison between SOTA L2Os with/without our proposed techniques (i.e., log loss over unseen optimizees with learned L2O v.s. training iterations of optimizees). Each curve is the average of ten runs.

In this section, we validate that the power of our proposed techniques can extend to improving previous SOTA L2O methods. CL and IL are applied to L2O-Scale and RNNprop, leading to corresponding strengthened models, named L2O-Scale-CI-IL and RNNprop-CI-IL, respectively[3].

As shown in Figure 3, our training techniques also immediately improve the two L2O models on all five optimizees, in terms of both objective value and the convergence stability.

## 5.3 Ablation study of our proposed techniques

For simplicity, we take L2O-DM as an illustrative example in the ablation case study. More results and details about our case study are referred to Appendix A1.

**Exploration and Exploitation** As shown in section 3.1, we can better balance exploration and exploitation by training L2O with more instances and longer for each instance. We set the number of training/validation steps to 100, 200, 500, 1000 respectively. We pick the model with the lowest validation loss during each 1,000 epochs so that we save 10 models for each experiment. We then evaluate the models by running these learned optimizers on MLP-orig optimizees while extending to

$10,000$ training steps. The average performance of 20 independent runs with different random seeds are reported, and the error bars are collected in the Appendix A1.

From the results in Figure 4, we gain the following two observations:

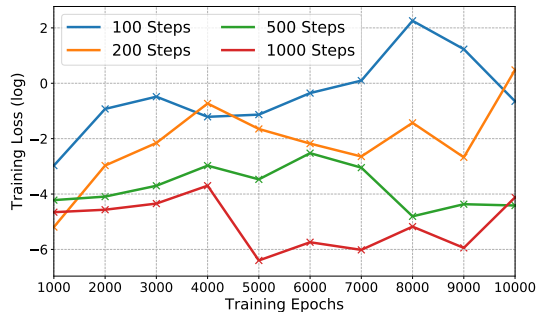

- With extra training epochs, the evaluation curve diverges, or ends up with larger losses. It suggests that redundant training epochs incur overfitting since it may amplify the bias on certain unrolling length ranges.
- The evaluation curves are less likely to diverge or degrade when increasing training epochs with more training optimization steps. The possible explanation is that longer exploitation helps with a more thorough landscape exploration and hence alleviates overfitting.

Figure 4: Evaluation performance of L2O (training loss of the MLP-orig) with different training settings. Each $\times$ mark on the curves represents the evaluated optimizee training loss after $10,000$ steps, guided by the corresponding L2O.

The best L2O model in Figure 4, training $5,000$ epochs with optimization step size $1,000$, is adopted as a strong augmented baseline (L2O-DM-AUG) for next-step comparisons.

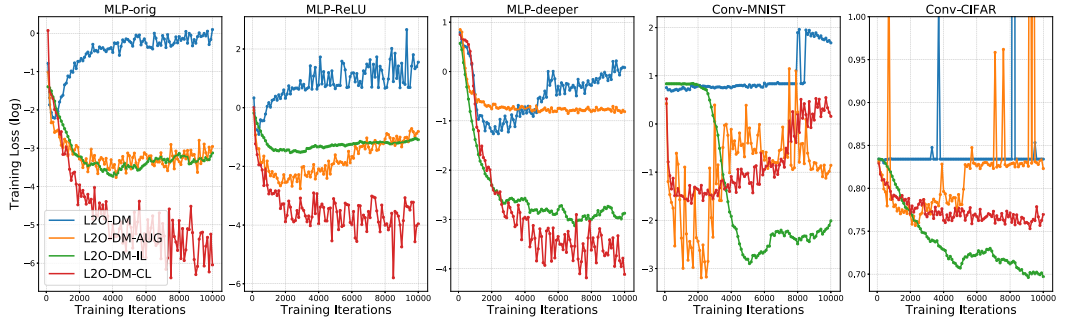

Figure 5: Evaluation performance of different L2O-DM variants (i.e., log loss over unseen optimizees with learned L2O v.s. training iterations of optimizees). Curves are the average of ten runs.

**Curriculum Learning**  We then assess the proposed curriculum learning [27] as in section 3.2. As shown in Figure 5, L2O-DM-CL surpasses L2O-DM-AUG with a significant performance margin, while only needs less than 1/14 training iterations of L2O-DM-AUG, thanks to its strategical focus on simpler training cases first. More details of CL and its cost analysis are referred to the Appendix.

**Imitation Learning**  We introduce off-policy imitation learning (IL) of analytical optimizers (Adam, SGD and Adagrad) as described in section 4 on L2O-DM. As shown in Figure 5, we observe that L2O-DM-IL converge better, especially on MLP-ReLU, Conv-MNIST and Conv-CIFAR where even L2O-DM-AUG tends to diverge. It suggests that optimization behaviors imitated from the analytical optimizers are in general beneficial, which adds extra effectiveness on top of CL as indicated by Figure 2 and 5. Besides, IL improves the stability of optimizee training. Appendix offers more analysis and more detailed settings for IL.

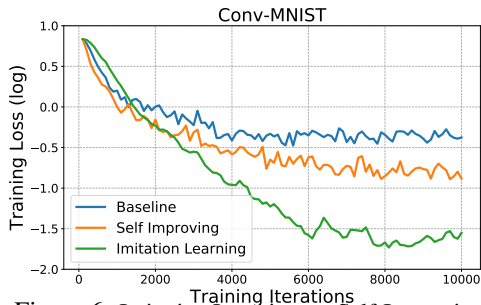

Figure 6: Imitation Learning v.s Self-Improving

**Imitation Learning versus Curriculum Learning**  We compare the performance across L2O-DM-CL-IL, L2O-DM-CL and L2O-DM-IL, as presented in Figures 2 and 5. We observe: i) On MLP optimizees, curriculum learning (CL) improves more than imitation learning (IL); On CNN optimizees, IL contributes more to the performance gain. Here is a possible explanation. The

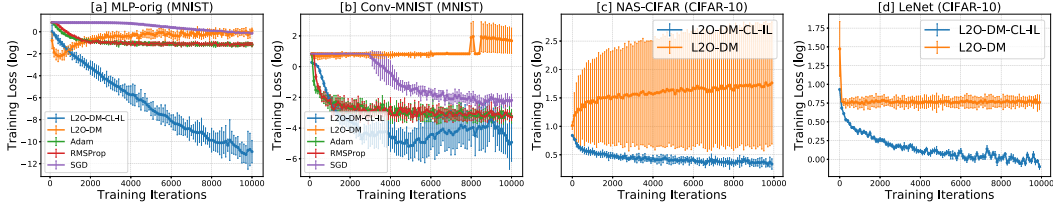

Figure 7: Evaluation results are collected (i.e., log loss over unseen optimizees with learned L2O v.s. training iterations of optimizees). [a]/[b] are comparisons against analytical optimizers. [c]/[d] present the transferability of learned L2O on complex optimizees. Curves with one standard deviation errorbar, are the average of ten runs.

curriculum learning (CL) technique mainly helps alleviate the notorious L2O truncation bias, while the imitation learning (IL) approach mainly helps L2O refer to the generally applicable optimization rules and hence avoid overfitting specific optimizee structures. Thus, IL appears to be a main contributor for L2O generalizing across different optimizees. ii) Combining CL and IL techniques always enjoys extra performance boost, compared to using either alone.

**Imitation Learning versus Self-Improving**   We also compare our proposed imitation learning regularizer with the self-improving techniques [38] which also take Adam, SGD, and Adagrad as their auxiliary optimizers to mix with. The probability of choosing L2O gradually grows to 1, while other probabilities linearly decay from 0.33 to 0 in 100 epochs of optimzier training. Figure 6 reports the previous SOTA L2O, L2O-Scale, as the baseline, and apply the two techniques (IL, and self-improving) on the top of it. Results demonstrate that while self-improving also demonstrates to be helpful in our case, IL shows to be certainly superior. We believe that the key reason lies in the *long-term coherency* of optimization trajectories. The imitation learning technique allows L2O to learn from the entire trajectories, with hand-crafted optimizers serving as end-to-end guidance. In comparison, self-improving breaks each optimization trajectory into mixed local "pieces" of applying either analytical or learned optimizers, therefore restricting learned optimizers to capturing only the dependency within local segments (e.g., a few iterations).

**Comparison against Hand-designed Optimizers**   We also conduct comparison against hand-designed optimizers on MLP-orig and Conv-MNIST optimizees with MNIST dataset, as presented in Figure 7 [a] and [b]. Plugging our proposed enhanced training techniques immediately boosts L2O to outperform analytical optimizers (i.e., Adam, RMSProp, and SGD) whose hyperparameters have been optimally tuned via a grid search, while the vanillan L2O-DM [1] quickly collapses and diverges in the same setting.

**Effectiveness on Complex Optimizees**   To show the effectiveness of our proposal on complex optimizees, we consider a non-conventional architecture sampled from a neural architecture search benchmark, and a larger LeNet model [42]), both on the CIFAR-10 dataset. The former, termed as NAS-CIFAR, is taken from the popular NAS-Bench-201 search space [43]. It[4] consists of multiple skip connections, convolutions, and average pooling operations, which is significantly more complicated than previously used simple instances of MLPs and CNNs. As shown in Figure 7 [c], our proposed techniques enable L2O trained on single-layer MLPs to generalize robustly to the much more sophisticated NAS-CIFAR, where vanillan L2O fails. We further validate our proposed training techniques can also successfully scale up to the LeNet case, as evidenced in Figure 7 [d].

## 6   Conclusion

Learning to optimize (L2O) is a promising field of meta learning that has so far been a bit held back by unstable L2O training and the poor generalization of learned optimizers. This work provides practical solutions to push this field forward. We propose a set of improved training techniques to unleash the great potential of L2O models. We apply our techniques to existing state-of-the-art L2O methods and consistently obtain performance boosts on a number of tasks. The contributions made in this work are of practical nature; we hope them to lay a solid and fair evaluation ground by offering strong baselines for the L2O community.

## Broader Impact

This work mainly contributes to AutoML in the aspect of discovering better learning rules or optimization algorithms from data. As a fundamental technique, it seems to pose no substantial societal risk. This paper proposes several improved training techniques to tackle the dilemma of training instability and poor generalization in learned optimizers. In general, learning to optimize (L2O) prevents laborious problem-specific optimizer design, and potentially can largely reduce the cost (including time, energy and expense) of model training or tuning hyperparameters.

## Footnotes

[1]In self improving, the loss for training L2O is still $\mathcal{L}_f(\boldsymbol{\phi}) = \sum_{t=1}^{N_{\text{train}}} \omega_t f(\boldsymbol{\theta}_t)$. The difference, compared with standard L2O training [1], is that here the optimizee $f(\boldsymbol{\theta}_t)$ may also be updated by the auxiliary optimizer, instead of just by the learned optimizer, at any step.

[2]This is the original setting [8] produced by meta training on its default ensemble of representative problems.

[3]We did not apply our technique to training L2O-Scale-Meta due to its complicacy, and also that our other simpler models with improved training already outperformed it.

[4]The detailed Architecture follows the demo one in `https://github.com/D-X-Y/NAS-Bench-201`: |nor_conv_3x3~0|+|nor_conv_3x3~0|avg_pool_3x3~1|+|skip_connect~0|nor_conv _3x3~1|skip_connect~2|.

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
