[Supplementary Material]

# Supplementary Materials: Training Stronger Baselines for Learning to Optimize

**Tianlong Chen[1*], Weiyi Zhang[2*], Jingyang Zhou[3], Shiyu Chang[4],**
**Sijia Liu[4], Lisa Amini[4], Zhangyang Wang[1]**
[1]University of Texas at Austin, [2]Shanghai Jiao Tong University,
[3]University of Science and Technology of China, [4]MIT-IBM Watson AI Lab, IBM Research
{tianlong.chen,atlaswang}@utexas.edu,{weiyi.zhang2307,djycn1996}@gmail.com
{shiyu.chang,sijia.liu,lisa.amini}@ibm.com

## A1 More Ablation Study

### A1.1 Exploration and Exploitation

As shown in Figure A1, The average performance of 20 independent runs with different random seeds are reported, and the error bars are also collected.

Figure A1: Evaluation performance of L2O (training loss of the MLP-orig optimizee) with different training settings. Each $\times$ mark on the curves represents the converge optimizee training loss after $10,000$ steps, guided by the corresponding L2O.

### A1.2 Curriculum Learning

L2O-DM-CL donates the enhanced L2O-DM with our proposed curriculum learning technique. We adopt $N_{train} = [100, 200, 500, 1000, 1500, 2000, 2500, 3000, ...]$, $N_{period} = 3$, $T_{period} = 100$. The original baseline in [1], L2O-DM, is trained with $N_{train} = N_{valid} = 100$. The strongest augmented baseline reported in the main text, L2O-DM-AUG, is trained with $N_{train} = 1000$, and $N_{valid} = 1500$. All learnable optimizers are trained with 5000 epochs.

**Results and Analysis** The results are presented in figure A2. We observe that the model trained by curriculum learning outperforms the two baselines (i.e., L2O-DM and L2O-DM-AUG) with fewer training iterations. For example, L2O-DM-AUG is trained for $5,000 \times 1,000 = 5,000,000$

Figure A2: Evaluation performance of different L2O-DM variants. Curves are the average of ten runs.

Figure A3: Evaluation performance of our enhanced L2O and previous SOTAs (i.e., log training loss v.s. training iterations of optimizees). Each curve is the average of ten runs.

iterations while L2O-DM-CL is only trained for $(100 + 200 + 500 + 1,000) \times 300 = 360,000$ iterations (the best model is saved during $\text{N}_{\text{train}} = 500$). It demonstrates that L2O-DM-CL surpasses L2O-DM-AUG with a significant performance margin, while only needs less than $1/14$ training iterations of L2O-DM-AUG, thanks to its strategical focus on simpler training cases first.

From the view of the potential advantages of curriculum learning as hypothesized in [2], the easy training examples represent a smoother global picture of the objective landscape. Therefore, the learner can avoid being trapped in local minimum at the beginning and converges to a better minimum. Furthermore, learning from a smoother objective landscape can also reduce the time wasted by noise examples and thus accelerate the convergence. In our specific case of learning to optimize, a training example input is a trajectory of optimizee parameters with corresponding gradients and updates. Smaller $\text{N}_{\text{train}}$ means shorter trajectories. As a result, the starting training example inputs are in the neighborhood of the initialization distribution of the optimizee parameters. Longer trajectories will cover more complex landscapes. Besides, a longer trajectory means longer horizontal dependency for the adaptive optimizer to learn. In addition, our proposed algorithm also provides a stop criterion. We stop when training on a new larger $\text{N}_{\text{train}}$ does not bring further decrease on the validation loss within $\text{N}_{\text{period}}$ periods, which indicates the L2O has been saturated with optimization knowledge.

### A1.3 Imitation Learning

L2O-DM-IL donates the enhanced L2O-DM with our proposed imitation learning regularizer, mimicking the optimization behaviors of analytical optimizers. We choose $(\mathcal{O}_i)_{i \in I}$ as the ensemble of Adam, SGD, and Adagrad. The hyperparameters of Adam $\beta_1, \beta_2$ are set to the default value $0.9$ and $0.999$ in Tensorflow. The learning rate of Adam, SGD, Adagrad, and the multi-task ratio $r$ are chosen by grid search as $0.01$, $0.01$, $0.01$, and $0.3$, respectively. The $\omega_t$ in equation (1) are set to 1. We train and validate both by $100$ optimization steps with $5,000$ epochs.

**Results and Analysis**  The results are shown in figure A2. we observe that L2O-DM-IL converge better, especially on MLP-ReLU, Conv-MNIST, and Conv-CIFAR where even L2O-DM-AUG tends to diverge. The multi-task learning of imitation of analytical optimizers forces the L2O to learn some general optimization policies, which improves the stability of optimizee training. According to [3], multi-task learning has the effect of regularization by inducing bias so that it helps alleviating

overfitting on random noise. Figure A3 also indicates that our proposed imitation learning technique has extra effectiveness on top of curriculum learning.

### A1.4   Training the L2O-DM baseline to surpass the state-of-the-arts

As shown in Figure A3, The average performance of 10 independent runs with different random seeds are reported, and the error bars are also collected.

## A2   More Training Details

We adopt the official training/testing split of MNIST and CIFAR-10 in our experiments.

- CIFAR-10 can be download at `https://www.cs.toronto.edu/~kriz/cifar.html`.
- MNIST is referred to `http://yann.lecun.com/exdb/mnist/`.
- All of our experiments are conducted on NVIDIA GTX 1080-Ti GPUs.