[Reviews · NeurIPS 2020]

Review 1

Summary and Contributions: This paper proposes changes to L2O to make a generic L2O algorithm easier and faster to train using common techniques like curriculum and imitation learning. They show their method give significant improvements across many existing evaluation criteria and methods. EDIT after rebuttal: Thank you very much for clarifying our concerns. I really appreciate running the additional experiments we requested and will increase my score. As an extra suggestion, please include the validation/test loss plots in your paper as opposed to training. It is hard to evaluate if the method is truly outperforming or just overfitting to a small training set. Also, for figure c/d on rebuttal, please include baseline optimizers (sgd/adam) too. I'm not concerned rather you beat those, but it would be helpful for those to be included as a reference.

Strengths: The work is technically sound, although the techniques used are not necessarily novel, the authors did a good job at evaluating a large combination of them which is non-trivial. This paper focuses specifically on improving a generic L2O algorithm rather than making the argument that their method is the best. I see this as a significant contribution that is very relevant to the NeurIPS community. Their empirical results indicate their method is a significant improvement to the SOTA of L2O which is extremely impactful to the NeurIPS community in the long run.

Weaknesses: Although the authors did a great job comparing against other L2O methods. I felt like there could be more comparison against hand-designed optimizers like ADAM/RMSProp/etc. It is hard to gauge how far we are from replacing the hard-coded ones with learned ones without a side-by-side comparison. I would also be interested on seeing performance across significantly different architectures (e.g. ones available in NAS search-spaces). rather than small changes in architectures like number of layers/stride/size. The MLP -> Conv comparison is helpful, but limited.

Correctness: As far as I could tell the experiments and claim are correct.

Clarity: Paper was well written overall although the explanation for Figure 4 could be more clear.

Relation to Prior Work: I think the paper did a good job discussing related work.

Reproducibility: Yes

Additional Feedback: I don't fully understand paragraph in line 108 on how it is related to explore-exploit in RL. I think we are not gaining much from this analogy. Minor typos/suggestions: Algorithm 1: switch from while loop to do-while to make sense of the "stop" flag. Or make while True and use continue/break statements L106: "optimzees" L219: "optmizee problemsduring" L272 "100, 200, 500, 1, 000" The comma for 1,000 is confusing


Review 2

Summary and Contributions: This paper presents simple techniques to improve the performance of L2O-DM, which is a standard baseline for learning to optimization problems. One is curriculum learning by controlling the number of training unrolling steps, and the other is imitation learning incorporating analytical optimizers. Although there are no theoretical foundations, empirical results support its usefulness to strengthen the baseline methods. Besides, the techniques can be incorpolated in the SOTA L2O models as well.

Strengths: (1) The proposed techniques are simple yet reasonable. (2) Empirical results are extensive and well support the benefit of imitation learning and curriculum learning in the L2O problem. (3) The paper is well written and easy to follow. (4) The topic is relevant to NeurIPS community.

Weaknesses: I don't have any major comments. Below are minor comments and questions. (1) Figure 5 suggests that the CL does not generalize well to the change of model architecture (MLP to CNN) compared to the IL. Can you explain the possible reason? (2) The definition of Lf(\phi), which is written in the line 98, is some what weird for me since the right side of expression does not depend on \phi.

Correctness: Empirical methodology are correct. Claims are validated by the empirical results.

Clarity: This paper is well written and easy to follow.

Relation to Prior Work: This paper is discuessed the difference with prior works. As I'm not an expert on the L2O literatures, I might miss some important literatures on this fields, especially regarding the technical proposal.

Reproducibility: Yes

Additional Feedback: == After Rebuttal == Thanks the authors for the rebuttal. I didn't find any additional concerns during the rebuttal. I'd like to keep my score.


Review 3

Summary and Contributions: This paper proposed a progressive training strategy for L2O models and aims to improve performance by mitigating the dilemma of truncation bias v.s. gradient explosion. Moreover, an off-policy imitation learning method has been introduced to prevent the L2O model from overfitting. Several experiments were conducted to demonstrate the effectiveness of the proposed method.

Strengths: 1. This paper provided a new aspect for the L2O model, which is novel and interesting. 2. Motivation and solutions are straightforward and reasonable. 3. The effectiveness of the proposed approach has been sufficiently demonstrated and detailed discussed with acquaint experimental results and corresponding analysis. 4. The code is available for better reproducibility.

Weaknesses: I have a criticism that most techniques are “adaptations” from RL literature. But that is just nitpicking: as they are new to L2O, and the authors explained the specific motivation to do those in L2O very well (section 3). The imitation learning also has a natural context for L2O, i.e., mimicking analytical optimization algorithms who are guaranteed to be good (section 4). Another suggestion: this paper was still (understandably) benchmarked on a few simple optimizes that were typically used by previous L2O literature (line 215, i - v). It remains questionable whether L2O, combined with proposed new training techniques, can eventually scale up to larger models, even just LeNet (2 conv + 3 fc) or so. Would be good to include some results or discussions here.

Correctness: Yes. Yes.

Clarity: This paper is well written and easy to read.

Relation to Prior Work: Yes. This paper focuses on a totally complementary direction to previous L2O literature.

Reproducibility: Yes

Additional Feedback:


Review 4

Summary and Contributions: this paper proposes a curriculum learning and imitation learning approach to improve the learning to optimize problem.

Strengths: 1. the method proposed is intuitive and practical, and the results seem promising 2. the paper tested the method on a range of tasks

Weaknesses: 1. the paper specifies difference from self-improving in section 4.2, but it's not clear why the proposed imitation learning is a better strategy 2. two strategies are proposed, but it is not clear from the experiments how much improvement comes from curriculum learning and how much comes from imitation learning

Correctness: yes it seems correct to me

Clarity: the paper is relatively clear, but there are some sections that are not well explained. it is not clear from the method section why certain design choices are made

Relation to Prior Work: it is discussed but i'm not an expert in the area, so I don't know if any work is missing.

Reproducibility: No

Additional Feedback:

[Author Response · NeurIPS 2020]

We genuinely appreciate all three reviewers' (#1,#2,#3,#4) valuable suggestions to strengthen our paper. We have
addressed all raised questions below: conducting new experiments to compare with hand-designed optimizers (#1)
and on more complex optimizee architectures (#1,#3); clarifying the experiment's observations (#2,#4); improving
the presentation (#1,#4) and fixing notations (#2). We will also fix all typos (#1) in our final version, and confirm our
promise to release our source codes and all trained models publicly, upon NeurIPS 2020 acceptance.

▷ Reviewer #1. **Q1. Comparison against hand-designed optimizers.** *Reply:* We follow your suggestions to conduct
comparison against hand-designed optimizers on MLP-orig and Conv-MNIST optimizees with MNIST dataset, as
presented in Figure 1 [a] and [b]. Plugging our proposed enhanced training techniques immediately boosts L2O
to outperform analytical optimizers (i.e., Adam, RMSProp, and SGD) whose hyperparameters have been optimally
tuned via a grid search, while the vanilla L2O-DM quickly collapses and diverges in the same setting. **Q2. Complex**
**optimizees (e.g. searched architecture from NAS).** *Reply:* We evaluate our enhanced L2O on a challenging optimizee,
NAS-CIFAR, from the popular NAS-Bench-201's search space [1]. NAS-CIFAR[1] consists of multiple skip connection,
convolution, and average pooling operations, which is significantly different with simple MLPs and CNNs. As shown
in Figure 1 [c], our proposed techniques enable L2O trained on single-layer MLP to generalize robustly to the much
more sophisticated NAS-CIFAR, where vanilla L2O fails. We further validate our proposed training techniques can
scale up to LeNet. More details are referred to Reviewer #3's Q2. **Q3. Readability of figure 4 and line 108.** *Reply:*
We sincerely appreciate your suggestion and will revise the caption in figure 4 for better readability. The analogy in line
108 is to unify our enhanced training framework under an RL view, which facilitates us to introduce core concepts like
the exploration-exploitation balance and imitation learning. We will make sure to picture this more clearly in our final
version. **Q4. Typos and Organization.** *Reply:* We will follow your suggestion to use the continue/break statement, fix
all typos, and remove the confusing comma in line 272.

▷ Reviewer #2. **Q1. IL generalizes better than CL from MLP to CNN.** *Reply:* It is a great observation. The
curriculum learning (CL) technique mainly helps alleviate the notorious L2O truncation bias, while the imitation
learning (IL) approach mainly helps L2O refer to the generally applicable optimization rules and hence avoid overfitting
specific optimizee structures. Thus, IL appears to be a main contributor for L2O generalizing across different optimizees.
In addition, we note that L2O equipped with both IL and CL achieves the best performance, as shown in figures 2 and 5.
**Q2. Notation in line 98.** *Reply:* Thank you. We will change the formulation $f(\boldsymbol{\theta}_t)$ to $f(\boldsymbol{\theta}_t, \boldsymbol{\phi})$.

▷ Reviewer #3. **Q1. Connections with RL.** *Reply:* We acknowledge Reviewer #3's comments and agree that our
proposed techniques his rooted in RL literature, but is uniquely motivated by and suitable for L2O. **Q2. Apply L2O on**
**LeNet.** *Reply:* We verify that our techniques can scale up the full LeNet: see Figure 1 [d].

▷ Reviewer #4. **Q1. Why imitation learning is superior to self-improving.** *Reply:* Great question. We believe that
the key point lies in the *long-term coherency* of optimization trajectories. The imitation learning technique allows
L2O to learn from the entire trajectories, with hand-crafted optimizers serving as end-to-end guidance. In comparison,
self-improving breaks each optimization trajectory into mixed local "pieces" of applying either analytical or learned
optimizers, therefore restricting learned optimizers to capturing only the dependency within local segments (e.g., a few
iterations). Experimental comparison details are referred to section 5.3. **Q2. Ablation of proposed strategies.** *Reply:*
In fact, we already compared the performance across L2O-DM-CL-IL, L2O-DM-CL and L2O-DM-IL, as presented in
figures 2 and 5 (we will add the curve of L2O-DM-CL-IL to figure 5 for better visibility). We observe: i) On MLP
optimizees, curriculum learning (CL) improves more than imitation learning (IL); On CNN optimizees, IL contributes
more to the performance gain (please also check our explanation in answering Reviewer #2's Q1). ii) Combining
CL and IL techniques always enjoys extra performance boost, compared to using either alone. **Q3. Readability of**
**the method section.** *Reply:* We will revise our method's presentation to draw a tighter connection with our design
philosophy of proposed techniques. For example, the above answer to Q1 will be integrated when we introduce IL.

Figure 1: **Ten runs.** [a]/[b] are comparisons against analytical optimizers (@R#1). [c]/[d] are complex optimizees (@R#1,@R#3).

[1] Xuanyi Dong and Yi Yang. Nas-bench-201: Extending the scope of reproducible neural architecture search. In ICLR, 2020.

## Footnotes

[1]Detailed Architecture: |nor_conv_3x3∼0|+|nor_conv_3x3∼0|avg_pool_3x3∼1|+|skip_connect∼0|nor_conv_3x3∼1|skip_connect∼2|, the demo architecture in [1]'s GitHub repository `https://github.com/D-X-Y/NAS-Bench-201`.


[Meta-Review · NeurIPS 2020]

This is a strong paper with sound fundamentals, and while the techniques used are not necessarily novel, the authors compensated throuugh a broad set of experiments, which will be helpful to a reader. The effectiveness of the proposed approach has been sufficiently demonstrated and code is made available for better reproducibility. One point of clarification that would help the reader is how much improvement comes from curriculum learning and how much comes from imitation learning.